# Incidental Findings in Study Participants: What Is the Researcher’s Obligation?

**DOI:** 10.3390/genes13101702

**Published:** 2022-09-22

**Authors:** Donna Schaare, Linda D. Ward, Luigi Boccuto

**Affiliations:** PhD Program in Healthcare Genetics and Genomics, School of Nursing, College of Behavioral, Social and Health Sciences, Clemson University, Clemson, SC 29631, USA

**Keywords:** incidental findings, research, ethics, policy, genomic testing

## Abstract

Background: As technology advances and genomic testing becomes commonplace, incidental findings, or the discovery of unrelated results, have increased. The American College of Genetics and Genomics (ACMG) established recommendations for the return of pathologic variants in 78 genes in the clinical setting based on medically actionable conditions from genes linked with preventable or treatable diseases. However, the lack of policy in the research setting poses a serious ethical dilemma for researchers, potentially threatening the participant’s trust and willingness to contribute to a process with more significant risk than benefit. Purpose: Our goal was to determine the preferred ethical approach to handling incidental research findings and suggest a new standard for investigators and participants. Methods: By employing Wueste’s IAJD Framework of ethical evaluation, the current research policy, as well as a proposed policy, were analyzed, and then a policy analysis was employed to ascertain feasibility. Results and Discussion: The current policy of leaving the decision of returning incidental findings up to the researcher’s discretion is an ethical failure from the consequential, deontological, and intellectual freedom perspectives. However, the proposed policy of implementing the ACMG guidance for researchers to satisfy ethical demands reinforces its moral fortitude. In a period of increasing public awareness, the community, which is the prospective research pool, has increased demands for autonomy and less paternalistic behavior from medicine and science. This paper synthesizes recommendations by numerous organizations to establish a mutually beneficial policy that will ensure the U.S. Department of Health and Human Services (HHS) goal, stated in the 2014 Joint Rule, of making participants “partners” in research a reality.

## 1. Background

As science and technology advance, genomic sequencing is becoming a standard tool clinicians and researchers use. The consequence is large amounts of data that can expose diagnostically unrelated results or “incidental findings” [1,2]. Incidental Findings (IFs), in the context of research, “are findings concerning an individual research participant that has potential health or reproductive importance and is discovered in the course of conducting research but is beyond the aims of the study” [3] (p. 1). Semantically and technically, incidental findings differ from secondary findings, another term often employed concerning unintended study results. Secondary findings, although not the primary target, are still linked to the research, while incidental findings fall outside the scope of the investigation. Currently, health policy is absent with regard to incidental findings and the investigator’s responsibility to the participant since they lack a clearly defined relationship like that between clinician and patient [4]. Specifically, this deficiency is apparent in the recent policy update of 2022, issued by The American College of Medical Genetics and Genomics (ACMG). It recommended the return of incidental findings involving 78 genes associated with medically actionable conditions to patients [5,6,7]. However, the ACMG recommendations are directed to clinicians alone and are, as stated in the 2021 update, “completely voluntary” [6]. Today, these findings represent a significant ethical dilemma for researchers. A recent example of this quandary is seen in the study by Nilsson et al. [8]. Identifying patients harboring germline *BRCA1* or *BRCA2* pathogenic variants became a secondary finding as a result of genomic screening affiliated with a cancer diagnosis. These gene variants, identified by the ACMG recommendations, meet the “criteria of analytic validity, clinical significance, and actionability (ACA),” the determining factors classified by broad consensus to alert individuals [8] (p. 202). Due to this recommendation in the clinical setting, the investigators felt it unethical not to notify the research participants of these significant findings for their future well-being. They not only re-contacted the participants but also, through semi-structured interviews, sought feedback to accumulate empirical evidence that reporting incidental findings was important to research subjects. So, with the ACMG guidance for health practitioners, the question remains; should researchers be required to inform research participants of incidental findings that include pathogenic variants of the 78 identified genes in the ACMG Recommendations?

Due to the migration from panel testing to genomic sequencing, the issues surrounding the return of incidental findings have been at the forefront of the socio-political-legal debate. Still, little has been resolved, especially in the research arena. Throughout the literature, conflicting guidance exists not only from an ethical perspective but regarding regulatory statutes and policy. From a socio-political perspective, President Obama’s White House was the first to weigh in on the dilemma of incidental findings with the Presidential Commission for the Study of Bioethical Issues Report in 2012 [9] and then a more in-depth follow-up in 2013 [1]. The reports included recommendations from an ethical perspective around incidental findings management in the research setting, but no policy was established. At the same time, the ACMG released its policy for healthcare providers reviewed earlier.

Regarding regulatory recommendations, the Secretary’s Advisory Committee on Human Research Protections (SACHRP), from the US Department of Health and Human Services (HHS), held public committee meetings. Their goal was to review the regulatory and ethical issues stemming from changes to Health Insurance Portability and Accountability Act of 1996 (HIPAA) and Clinical Laboratory Improvement Amendment of 1988 (CLIA) in returning individual results to human research subjects. One of the topics covered in a summary statement released in 2017 was the return of incidental findings, an update from the initial 2015 report [10,11]. The committee emphasized the enormity of the situation and that “there is no single preferred model for making decisions about whether to return incidental findings” due to the ethical and legal complexity of the issue [11]. They continued recommending return of findings in certain circumstances when the findings were valid and actionable. However, non-CLIA-certified laboratories are responsible for research testing which makes validity questionable [11]. Concerning regulatory status, “the HHS and FDA regulations (45 CFR 46 and 21 CFR 50 and 56, respectively) are silent regarding the return of incidental findings to research subjects, and neither directly require nor disallow this activity” [11]. In their document, *Investigator Responsibilities-Protecting the Rights, Safety, and Welfare of Study Subjects,* the U.S. Food and Drug Administration (FDA) advocates for investigators to inform participants to seek medical care for a condition or illness uncovered unrelated to the study [11].

From the research participant’s perspective, the regulation included in HIPAA’s Privacy Rule §164.524 guarantees individuals’ access rights to Protected Health Information (PHI), which should include incidental findings, if requested, with some limitations regarding Randomized Controlled Trials (RCTs) and blinding [10]. Yet, it is not that straightforward. The CLIA regulations that fall under HIPAA apply only to CLIA-certified labs, which, as stated earlier, do not pertain to research laboratories because they do not usually report patient-specific results [10]. Even more confusing is the Center for Medicare and Medicaid Services (CMS) position regarding non-CLIA certified labs returning test results to patients as violating their research exemption [10]. This regulation makes it very difficult for research labs to comply with HIPAA regulations. To date, however, no court cases have considered the researcher’s liability for not returning incidental findings [1].

In the current socio-political environment, it is apparent why returning incidental findings to research participants is confusing for all. In this case, a lack of a policy is the existing policy; the result is leaving the decision up to the researcher’s discretion (Table 1). This lack of policy benefits the researcher because, without a duty to inform, the researcher is free to do what is convenient to accomplish their aims and to improve generalizable knowledge without obligation to the individual participant. However, the limitation to this model lies with the research subject who is the one who will incur the risk with no individual benefit or reward except altruism. This archetype is not sustainable if advertised to the public.

## 2. Identify

In order to analyze the ethical challenges of a policy recommending investigators to return incidental findings, consisting of the 78 medically actionable genes from the ACMG guidance to research participants, Wueste’s four-step IAJD Framework: Identify, Analyze, Justify, and Decide will be applied. The analysis begins with the identification of the key stakeholders in this ethical and policy debate [12]. The research participant and investigator have much to gain or lose in this situation, in addition to the participant’s family and the research laboratory. Some other groups that have a stake in the outcome include grant providers, and payers, both the government and private insurance.

The issues for the research participants and their families are extensive, including the implications for future health. The psychological aspects of returning genetic findings can include fear, guilt, and even denial, as the participant worries about what medical complications may result for themselves or their loved ones [8]. Often, patients feel to blame when they pass their genetics to their offspring. The knowledge of the incidental findings may also place financial stress on the participant and their genetic relations as they potentially pursue further testing and education to understand the disorder. The burden of meeting with a genetic counselor, who may not be convenient or readily available, also plays a part in this situation. However, not knowing of a medically actionable condition may be far more damaging, not only from a health but also from a psychological and financial perspective [8,13].

The researcher’s concerns around returning incidental findings involve time and resources, which are very limited in the research setting [1]. Understanding and interpreting incidental findings may require hiring a genetic expert. Additionally, contacting, educating, and recommending clinical follow-up for the participant will be time-consuming and potentially costly. Therefore, returning incidental findings would also impact the grant funding institutions requiring larger grants to accomplish the same research goals.

The research laboratory may be in the most challenging situation with the legal requirement to comply with HIPAA’s Privacy Rule and the CMS guidance that non-CLIA certified labs should not “under any circumstances” release test results to test subjects [10]. Since both regulations, HIPAA and CMS, directly oppose each other, either option may be a losing proposition for them.

Finally, the payers, government, and private insurance will be concerned with financial stress to the system since resources are limited in healthcare. The payer must cover confirmatory testing and genetic counseling costs that may be incurred when incidental findings are reported to the research participant. Any prophylactic surgeries or increased screening protocols may be expensive, but cancer treatment or critical care can also be costly. Usually, preventative care tends to create less financial burden than handling an existing condition.

## 3. Analyze

Policy analysis begins with applying certain ethical principles to each stakeholder’s issue. Although Belmont’s/Beauchamp-Childress’ Principlist Approach is understood to encompass most aspects of health ethics, some ethicists have noted the need for additional principles due to the bioethical issue under scrutiny for a more comprehensive evaluation [14]. Therefore, by adding Intellectual Freedom and Responsibility Principle, introduced by the Presidential Commission for the Study of Bioethical Issues in their report from 2013, to the Principlist Approach, the behaviors displayed by researchers are subject to openness and transparency [1]. These are the ethical principles to be explored for the policy analysis (Table 2).

### 3.1. Consequential: Beneficence and Nonmaleficence

From the consequential perspective, by applying the principle of beneficence to the participant, the researcher, in coordination with the Intitutional Review Board (IRB), needs to promote well-being. Therefore, any finding that is medically actionable, clinically valid, and may “help forestall or prevent harm” must be returned for the benefits to outweigh the risks [4]. From the opposite standpoint, divulging findings that can cause distress without any clinical value, e.g., variants outside of the 78 genes included in the ACMG Guidelines may inflict harm and fail the nonmaleficence provision [1,3]. The participant and family may suffer some negative consequences of knowing of a pathogenic genetic condition, such as the emotional and financial burden of seeking out additional testing, meeting with a genetic counselor for education, and potential preventive medical treatment. An overall health benefit may outweigh these consequences. As an extension of the investigator, the laboratory also needs to safeguard the welfare of the research participant by providing results when the benefits outweigh the risks, as in the ACMG recommendations. From the payer perspective, prevention should ideally be more cost-effective than treatment. Therefore, disclosing actionable incidental findings, although initially expensive in resource utilization, may reduce overall health expenses. Returning results is not the easy option of ignoring the findings that the current policy allows; however, it is the more ethical choice from a consequential perspective.

### 3.2. Deontological: Autonomy/Respect for Persons

Regarding the deontological principle of autonomy and respect for persons, when the researcher reveals actionable incidental findings, the research subject is being treated with respect by becoming better informed and equipped to make future medical decisions. Their health decisions are firmly in their hands as well as that of their family. Additionally, with the current policy, the participant is being treated as a means to an end. By not reporting the findings, the investigator is free to accomplish their goal alone, not having to incur the added time and expense of notifying, educating, and following up on results not pertinent to their study aims. By returning results, the lab, as an investigation agency, is also fostering the participant’s autonomy at the potential risk of their research status by violating CMS guidance and jeopardizing their research status. Again, the deontological perspective favors the more complicated process of returning incidental findings.

### 3.3. Intellectual Freedom and Responsibility: Progress and Accountability

Returning incidental findings will require the investigator to invest more time and resources in their participants but will translate to a heightened appearance of genuine concern for research subjects. This return will foster the trust needed by researchers from participants to ensure continued investigation and increased generalizable knowledge. Today is the era of research resulting in increased technical knowledge, which ultimately translates into clinical practice and benefits the individual. Therefore, it is necessary to demonstrate beneficence or risk losing the ability to continue attracting research subjects and pursue scientific advancement for all [15]. By not disclosing serious medical findings, researchers may appear to be using research subjects as a means to an end. The subjects would begin to feel more like *M. musculus* than *Homo sapiens*, again lending weight to the positive ethical choice of requiring the return of incidental findings to participants.

### 3.4. Justice: Equity of Distribution

Finally, from the ethical perspective of justice or equitable distribution of resources, the alternative policy is negative for most stakeholders involved, unlike the current policy of “do not ask, do not tell.” Returning incidental findings will put an additional burden on the researcher, regarding time and finances, in an environment with minimal resources and a goal of societal good over individual benefit. As stated earlier, this is a predicament because, without the participation of individuals, generalizable knowledge is not attainable. However, this does not diminish the undue tax that distributing incidental findings will put on the researcher. Additionally, the lab potentially loses its CLIA-exempt status by supplying test results to individual participants unless CMS changes its mandate [10]. Payers, as stated earlier, would incur costs upfront but may save money in prevention versus treatment. From the justice perspective, the only stakeholder benefiting from the return of incidental findings is the research subject. They can use the medical resources they need for education, prevention, and treatment. However, knowing they have a medically actionable condition does not mean they are free from possible discrimination in securing non-GINA (Genetic Information Nondiscrimination Act of 2008) protected insurance, like life, disability, and long-term care [16,17].

## 4. Justify

As stated by Dr. Wueste in his IAJD Framework, “Convergence begets confidence”. Although not complete, the option to recommend the return of incidental findings as outlined in the ACMG recommendations to research participants is overall positive concerning the four ethical principles considered (Table 3) [12]. Unlike the current policy (Table 4), which is unfavorable from the consequential, deontological, and intellectual freedom/responsibility perspectives, convergence with the policy to extend ACMG recommendations to research testing is much more apparent. Justice seems to be the only place of disjunction due to the discordance between the rights of the individual and the greater societal good. With the public’s increasing genetic knowledge, a policy to return incidental findings must be established soon, or investigators risk losing the significant benefits research produces due to lack of participants, which the Belmont Report forewarned [15]. An additional example of the sustainability of returning incidental findings is from Haukkala et al. [13], a biobank study, where 27 subjects were identified by investigators to harbor variants in genes, *KCNQ1* and *KCNH2,* associated with long QT syndrome (LQTS). Similar to the Nilsson et al. [8] study, patients were re-contacted, re-tested at a CLIA-certified lab, educated, and then surveyed to ascertain the demand by the study participants. Again, the results were positive overall, and all 17 of the 27 responders believed that incidental findings should be delivered.

## 5. Decide

Therefore, since agreement for 75% of the ethical principles was attained for the proposed policy of returning incidental findings to study participants, the decision to move forward with a policy change is indicated. Instead of leaving the decision to disclose incidental findings up to the discretion of the investigator, a new policy that recommends researchers inform research participants of incidental findings that include the 78 identified genes, as outlined in the ACMG Recommendations governing clinicians, should be established. From a socio-political-legal perspective, multiple contextual factors influence the need to make this change, including situational, structural, cultural, and international dynamics [18]. The COVID-19 pandemic represents a situation that has changed the dynamic of how research is conducted, from randomized controlled trials to real-world evidence. The global population has become research participants, and there is an increased focus on understanding the generalizable knowledge process shared in the social media culture. Furthermore, health disparities impact the accuracy of study data produced due to the lack of inclusion of certain ancestral groups [19]. Additionally, there is a new perspective that it is one’s duty to help society by individually contributing genetic information to national and international programs like All of Us and the UK Biobank [20]. Finally, even before the COVID-19 pandemic, the HHS stated a change in the relationship of participants to research in the Preamble of the 2014 Joint Rule. The goal was to make participants “partners” in research, especially in precision medicine initiatives [10]. All these factors, plus a global push to reduce paternalistic behavior in medicine and increase transparency in research studies, have brought the international law of human rights to the forefront of socio-political-legal discussion and have set the stage for policy change in the research setting [2,3,16,20].

## 6. Proposed Policy Implementation

Before implementing this new policy, standards must be established reflecting the ethical principles previously discussed. Some suggestions are alluded to in multiple documents, including *Anticipate and Communicate: Ethical Management of Incidental and Secondary Findings in the Clinical, Research, and Direct-to-Consumer Contexts*, as well as the SACHRP Recommendations Attachment C and F. Standards development will require interdisciplinary committees to be established with representatives from the various stakeholder groups, including researchers, potential participants, labs, clinicians, payers, advocacy groups, academia, industry, and the government. The goal will be to reach a consensus around the imperatives needed to be included in the standards and the resources necessary to execute the policy and stay within the ethical parameters discussed [21].

Some suggestions these committees may want to consider come from the recommendations by the Presidential Committee and the SACHRP (Table 5). The first consideration concerns informed consent, which needs to include a detailed plan regarding the potential uncovering of incidental findings and whether they will be returned based on feasibility. The implications of germline testing need to be explained to the participant, including how the findings may impact their own lives and their extended family [22]. The ethical responsibility to inform their family must also be stressed [22,23]. As stated earlier, only medically actionable pathogenic variants should be returned following the recommendation of the ACMG [24]. An example of an exemption from this policy is de-identified data with no way to reverse the process in a biobank, or genome sequencing without complete examination of findings, focusing only on specific genes or regions. If that were the case, then the investigator must notify the potential subject so they have the choice to decline to participate, which is currently not a standard option in the consenting process.

Additionally, the investigator should exclude any participant who does not want their incidental findings returned as part of the exclusion criteria since this is another topic of controversy. That factor must take into consideration the type of informed consent employed. Multiple models are available including, “traditional” or pre-study; “staged”, consent in phases, “mandatory”, forced consent, or “outsourced”, consent through a third-party [25]. The National Heart, Lung, and Blood Institute (NHLBI) Working Group stated, in a publication from 2006, that researchers should even inform participants who have opted out [3]. By excluding them, autonomy is respected for any potential research subject, which would require a traditional pre-study informed consent to be completed [4].

The next matter involves handling non-CLIA-certified labs, which include most research facilities. As suggested by the SACHRP, a researcher who identifies an ACMG genetic variant because of a research test in a non-CLIA-certified lab should be able to refer subjects to a CLIA-certified lab for testing. Additionally, this parameter needs to be clarified and ratified with regulatory respect to HHS [10].

The third category deals with expertise that may be lacking by the investigator. Using the guidelines suggested by the ACMG, there should be less uncertainty over the actionability of the findings. Furthermore, this protocol was endorsed by the National Academy of Sciences, Engineering, and Medicine in 2018 and the American Society of Human Genetics in 2019. However, gene-gene protein protein interactions, modifiers, and epigenetic factors can inhibit disease expression even in the presence of pathogenic variants in accordance with the inheritance pattern. These rapidly changing factors are essential for researchers to consider when presenting incidental findings. Therefore, it may be wise for researchers to employ a clinician or genetic specialist for consultation services when a variant is identified in one of the 78 genes mentioned by the ACMG [4].

Finally, to reduce the cost of educating and following up with the participant, it may be practical to require the disclosure of the research subject’s clinician during the informed consent process. Instead of presenting the incidental genetic findings to the participant without protein or metabolic equivalents, the researcher, in conjunction with the medical provider can act together, assuming the responsibilities of confirming findings, instructing, and monitoring the participant’s progress, instead of leaving them with many unknowns. This shared responsibility will require increased coordination and communication, skills that may require additional education for the researcher and clinician [4]. These are just some of the specifics that the new policy will need to address to overcome the ethical limitations of the current guidelines.

A recent example illustrating the implementation of these changes in returning incidental findings to participants was executed by Beil et al., in 2021 [26]. The investigators set up a protocol to identify and recontact participants of the Cardiovascular Health Improvement Project (CHIP) biobank with aortic disease who harbored pathogenic variants in 11 genes associated with aortopathy. Variants in 6 genes were identified (*COL3A1, FBN1, LOX, PRKG1, SMAD3,* and *TGFBR2*) in 26 participants. The researchers followed a five-step procedure that included validating the sample, alerting the participant, genetic counseling phone call followed by appointment, and a CLIA lab-validated genetic test. Additionally, the investigators assessed the psychological implications of returning these findings and the cost per participant to the research team. The results reinforced the importance of disclosing incidental findings and the authors reported, “participants were satisfied with the disclosure process, generally understood the meaning and implications of test results, and did not experience adverse psychological effects” [26] (p. 9). Although no cost-effectiveness studies have been performed, this investigation assessed the cost per participant (~$400) for the above five steps. The investigators determined that expenses can be contained when CLIA confirmatory analysis is specific to the variant and does not include whole genome sequencing [26]. These findings may be beneficial in assisting the creation of the new policy.

## 7. Conclusions

Without a transformation in the perceived relationship between investigators and participants, science risks losing the opportunity to continue to increase generalizable knowledge through the research setting. The climate is poised for change. As Dr. Wylie Burke stated in her presentation, *Partnering with Communities in Genomic Research,* “research is moving from a risk model to a respect model” [27]. Without demonstrating that respect, by returning medically actionable incidental findings, such as the 78 genes specified by the ACMG, future participants may not feel the overall risk and lack of autonomy, combined with the investigator’s lack of accountability, is worthy of their contribution. It is time for the stakeholders to come together and establish a mutually beneficial policy like the one proposed in this paper. Additionally, legal statutes must be added to HHS and FDA regulations 45 CFR 46 and 21 CFR 50 and 56, respectively, to clarify the researcher’s duty to inform and enforce these protections.

## Figures and Tables

**Table 1 genes-13-01702-t001:** Recommendations, Laws, and Acts around returning Incidental Findings in Research.

Title	Institution	Date
*Privacy and Progress in Whole Genome Sequencing*	Presidential Commission for the Study of Bioethical Issues	2012
*For Researchers: Incidental and Secondary Findings Primer*	Presidential Commission for the Study of Bioethical Issues	2013
*Anticipate and Communicate: Ethical Management of Incidental and Secondary Findings in the Clinical, Research, and Direct-to-Consumer Contexts*	Presidential Commission for the Study of Bioethical Issues	2013
*Attachment C: Return of Individual Results and Special Consideration of Issues Arising from Amendments of HIPAA and CLIA*	Protections (OHRP) Office for Human Research	2015
*Attachment F—Recommendations on Reporting Incidental Findings*	Protections (OHRP) Office for Human Research	2017

**Table 2 genes-13-01702-t002:** Ethical Principles.

Ethical Principles	Subjects	Description
Consequential	Beneficence Nonmaleficence	Ensure wellbeing Do no harm
Deontological	Autonomy	Respect for Persons
		Never used as a means to an end
Intellectual Freedom & Responsibility	Scientific Innovation and Accountability	To further scientific progress and Researcher answerable for actions
Justice	Equality	Equity based distribution
		of resources

**Table 3 genes-13-01702-t003:** Alternative Policy.

Ethical Perspectives	Participant/Family	Researcher	Lab	Payers	Overall POSITIVE
**Consequential** (Beneficence/Nonmaleficence)	Positive	Positive	Positive	Positive	Positive
**Deontological** (Autonomy/Respect for Persons)	Positive	Positive	+/−	N/A	Positive
**Intellectual Freedom** & **Responsibility** (Progress & Accountability)	Positive	+/−	N/A	N/A	Positive
**Justice** (Equity of Distribution)	+/−	Negative	Negative	+/−	Negative

**Table 4 genes-13-01702-t004:** Current Policy.

Ethical Perspectives	Participant/Family	Researcher	Lab	Payers	Overall NEGATIVE
**Consequential** (Beneficence/Nonmaleficence)	Negative	Negative	Negative	+/−	Negative
**Deontological** (Autonomy/Respect for Persons)	Negative	Negative	+/−	N/A	Negative
**Intellectual Freedom** **& Responsibility** (Progress & Accountability)	Negative	+/−	N/A	N/A	Negative
**Justice** (Equity of Distribution)	Negative	Positive	Positive	+/−	Positive

**Table 5 genes-13-01702-t005:** Policy Changes.

Policy Changes	Description
Informed Consent	Plan-Possibility of Incidental Findings
	Return or not based on feasibility
	Opt-in or out
Non-CLIA certified lab results	Refer to CLIA certified lab for confirmatory testing
	HHS clarify and ratify regulations
Consultation services of genetic specialist	78 genes-ACMG recommendation
Disclosure of participant clinician	Coordinate follow-up care

## Data Availability

Not applicable.

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
