# Peer review of "Incidental Findings in Study Participants: What Is the Researcher’s Obligation?"

_genes, 2022, doi:10.3390/genes13101702_

Round 1

Reviewer 1 Report

This publication raises a very interesting topic which is at the moment a hot issue in the clinical genetic. I have read the publication with great interest as a clinical genetics researcher. From my point of view, it raises many interesting ethical, financial and legal aspects that broaden the understanding of the problem of reporting incidental findings.

Author Response

Thank you so much for your positive feed back and encouragement! We appreciate your support!

Reviewer 2 Report

Comments for Authors

The issue that the researchers are dealing with is an extremely important and pending issue for both diagnostic and research laboratories in the genomic era.

In this study, authors have aimed to determine  the preferred ethical approach to handling incidental genomic research findings and suggest a new standard  for investigators and participants. The authors describe current policy that leaving it to the researcher to decide to return incidental findings to the participant, highlight an ethical failing. And also, they point that to synthesize recommendations from multiple organizations to create a mutually beneficial policy.

The article is well written in all aspects. However, a few issues that I am concerned about are summarized below;

As genetic analyzers moved from research labs to diagnostic labs, and as the number of genomic studies increased, many unknown aspects of the genome were and continue to emerge. There are many more unknowns...

1-Gene-gene protein-protein interactions, modifiers, and epigenetic factors can inhibit disease expression even in the presence of pathogenic variants in accordance with the inheritance pattern. The authors should note that researchers should be aware of this rapidly changing information when presenting incidental findings. Instead of presenting the incidental genetic findings to the participant without protein and metabolic equivalents, could it be suggested as a solution to include the participant in a follow-up system among researchers, clinicians and policy makers? Instead of leaving the participant alone with so many unknowns, knowing that this situation is followed by experts and stakeholders positively affects the participant and ensures sharing responsibility.

2-Are the authors aware of whether there is a cost-effectiveness study with a retrospective analysis of incidental genomic findings? If not, they can state it as a need.

Thanks for your hard work.

Author Response

Thank you so much for your positive feedback and encouragement. Your insightful and valuable suggestions are greatly appreciated. We will address each suggestion below and thank you again!

  1. We agree and added the suggestion of having the researcher work as a team with the participant's clinician to manage results and aid in shared responsibility (lines 350-353 and 358-362).
  2. Currently, there are no cost-effectiveness studies however, the Beil et al., 2021 study analyzed the cost of returning incidental findings to biobank participants, a very important consideration. Thank you! (lines 403-407)

Reviewer 3 Report

The article is on the whole interesting, deals with a very important public health issue and is structured in an effective and coherent way. 

I have some comments related to minor revisions which I hope will be useful to the authors:

- In the affiliations the city and the Country of the institutions of the Authors are missing.

- In the introduction the Authors cited Recommendations, Laws, Acts, it would be useful to the reader create a table contain the title and institutions of the documents cited.

- In the published literature we find incidental findings and also secondary findings, in the Authors? Opinion is there any differences in the management? In case, briefly described this aspects. 

-In the discussion, when the Authors describe  (pag7 line 278) the necessity to report in the informed consent form the question regards the management by researchers of the possible return results of actionable Incidental Findings, I suggest a little deepened of this aspect describing what it would be useful to  indicate in the consent form. 

In addition to the article cited by the Authors, I suggest some articles that can help in this:

Models of consent to return of incidental findings in genomic research. Appelbaum PS, Parens E, Waldman CR et al. Hastings Cent Rep. 2014 Jul-Aug;44(4):22-32. 

Return of genetic testing results in the era of whole-genome sequencing.

Knoppers BM, Zawati MH, Sénécal K.

Nat Rev Genet. 2015 Sep;16(9):553-9. 

Responsibility, identity, and genomic sequencing: A comparison of published recommendations and patient perspectives on accepting or declining incidental findings. Boardman F, Hale R. 

Mol Genet Genomic Med. 2018, 6:1079-1096. 

New Frontiers and Old Challenges: How to Manage Incidental Findings When Forensic Diagnosis Goes Beyond. 

Caenazzo L, Tozzo P, Dierickx K.

Diagnostics 2020, 10, 731; 

Author Response

Thank you so much for your positive feedback and encouragement! Your insightful and valuable suggestion are greatly appreciated. We will address each suggestion below.

  1. Thank you! We added city and country to affiliations.
  2. Thank you! We agree it may be easier for the reader to assess in table form. We created a new Table 1.
  3. We agree over the confusion around terminology. We clarified in lines 39-42.
  4. Thank you so much for the suggested readings. It helped clarify more aspects of the consent process. We included multiple aspects from your suggestions. Lines 328-341, 350-353, and 356 and References 22-26.

Reviewer 4 Report

This is a well structured manuscript of a very controversial and timely topic.

Minor comments:

The authors refer to 50+ ACMG genes, the most current update, the ACMG version 3.1 list now includes 78 genes, (updated in 2022), I'd suggest including the statement published in https://doi.org/10.1016/j.gim.2022.04.006 in addition to ref 5, 6.

It would improve by providing further  analysis of research where reporting incidental findings have been implemented in research pipelines  such as  :  Beil, A., Hornsby, W., Uhlmann, W.R. et al. Disclosure of clinically actionable genetic variants to thoracic aortic dissection biobank participants. BMC Med Genomics 14, 66 (2021). https://doi.org/10.1186/s12920-021-00902-5

Also, how those the ACMG guidelines compare to the European (or other societies) recommendations?

Author Response

Thank you so much for your positive feedback and encouragement. Your insightful and valuable suggestions are greatly appreciated. We will address how we handled each suggestion below.

  1. Thank you for providing us with the most current update. We added the reference and updated every ACMG recommendation to 78 throughout the paper. See reference 7.
  2. Thank you so much! It was a great paper to add to the proposed policy implementation as an example of ways to implement changes. Lines 382-408. Reference 26.
  3. We agree and added the National Academy of Sciences, Engineering, and Medicine as well as ASHG. Lines 348-349. Thank you! 
